# Seg-VAR: Image Segmentation with Visual Autoregressive Modeling

**Rongkun Zheng**[1]  **Lu Qi**[2]  **Xi Chen**[1]  **Yi Wang**[3,4]  **Kun Wang**[5]  **Hengshuang Zhao**[1†]

[1]The University of Hong Kong [2]Insta360 [3]Shanghai Artificial Intelligence Laboratory
[4]Shanghai Innovation Institute [5]SenseTime Research
{zrk22@connect, hszhao@cs}.hku.hk

## Abstract

While visual autoregressive modeling (VAR) strategies have shed light on image generation with the autoregressive models, their potential for segmentation, a task that requires precise low-level spatial perception, remains unexplored. Inspired by the multi-scale modeling of classic Mask2Former-based models, we propose Seg-VAR, a novel framework that rethinks segmentation as a conditional autoregressive mask generation problem. This is achieved by replacing the discriminative learning with the latent learning process. Specifically, our method incorporates three core components: (1) an image encoder generating latent priors from input images, (2) a spatial-aware seglat (a latent expression of segmentation mask) encoder that maps segmentation masks into discrete latent tokens using a location-sensitive color mapping to distinguish instances, and (3) a decoder reconstructing masks from these latents. A multi-stage training strategy is introduced: first learning seglat representations via image-seglat joint training, then refining latent transformations, and finally aligning image-encoder-derived latents with seglat distributions. Experiments show Seg-VAR outperforms previous discriminative and generative methods on various segmentation tasks and validation benchmarks. By framing segmentation as a sequential hierarchical prediction task, Seg-VAR opens new avenues for integrating autoregressive reasoning into spatial-aware vision systems.

## 1 Introduction

Image segmentation—the task of partitioning pixels into semantically meaningful regions—requires models to capture hierarchical spatial relationships, from coarse object categories to fine-grained instance boundaries. While advancements in convolutional and transformer-based architectures have pushed performance on semantic, instance, and panoptic segmentation, these approaches often treat segmentation as a parallel prediction task, struggling to model the iterative, context-dependent spatial and semantic relationships in complex scenarios. Recent work in visual autoregressive (VAR [58]) modeling, which sequences images into tokens for generative tasks, offers a promising alternative: its sequential, context-accumulating nature could naturally capture the progressive refinement inherent to segmentation. However, existing VAR frameworks prioritize image synthesis, neglecting their potential to unify segmentation tasks through structured spatial autoregression.

A key obstacle lies in representation: most autoregressive frameworks encode images into latent spaces that lack explicit spatial or instance-level structure. For example, while Generative Semantic Segmentation (GSS [5]) learns latent distributions to guide segmentation, its encoder fails to disambiguate overlapping instances or preserve fine-grained positional cues. Conversely, autoregressive image generators typically treat pixels or patches as unordered tokens, sacrificing the geometric

---

[†]Corresponding author.

39th Conference on Neural Information Processing Systems (NeurIPS 2025).

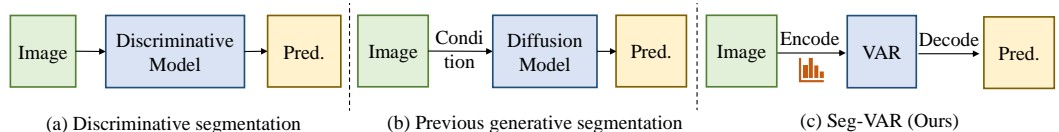

Figure 1: Our Seg-VAR is a visual autoregressive model that is designed for generic image segmentation. Different from (a) traditional discriminative segmentation models and (b) diffusion-based generative models that mainly take input image as a condition, our Seg-VAR rethinks segmentation as a sequence modeling task by encoding the input image to a latent distribution and generating the masks hierarchically.

control needed when generating images that demand strong spatial relationship, such as behind or next to. Bridging this gap requires a VAR framework that (1) decomposes images into hierarchical, position-aware tokens to represent objects at multiple scales, and (2) leverages autoregressive dependencies to propagate spatial coherence across these tokens.

In this work, we introduce Seg-VAR, a visual autoregressive model that is designed for generic image segmentation (semantic, instance, and panoptic). Seg-VAR is built on visual autoregressive (VAR) modeling and employs a hybrid design that combines hierarchical autoregressive decoding with next-scale prediction principles. As shown in Fig. 1, our approach rethinks segmentation as a conditional autoregressive mask generation task, where discrete tokens encode both semantic classes and instance-aware positional information. Our approach hinges on multiple innovations: 1) Spatial-aware seglat encoding: We introduce seglats—latent representations of segmentation masks—generated via a novel encoder that maps masks to discrete tokens using location-sensitive color mapping. This mechanism assigns unique RGB values to instances based on their spatial centroids, enabling transformers to distinguish overlapping objects through positional awareness. 2) Hierarchical autoregressive decoding: A transformer-based decoder reconstructs masks by sequentially predicting seglat tokens conditioned on image features, ensuring spatial coherence through autoregressive attention. This mimics human-like iterative refinement, where early tokens establish global context and later tokens resolve local ambiguities. 3) Multi-stage latent alignment: A three-stage training strategy first learns seglat-image correlations, refines latent transformations, and finally aligns image-derived priors with seglat distributions via KL divergence minimization.

By training SegVAR to maximize the likelihood of ground-truth token sequences—while jointly optimizing pixel-level mask fidelity—the model learns to harmonize semantic accuracy with geometric consistency. Experiments demonstrate state-of-the-art performance on COCO, Cityscapes, and ADE20K, with significant gains in occluded scenes and small-object segmentation. Notably, Seg-VAR's autoregressive tokenization generalizes across segmentation tasks: the same architecture achieves top-tier results in semantic, instance, and panoptic settings, showcasing VAR's versatility as a unified paradigm for spatial understanding. Our contributions are as follows:

- We analyze the limitations of existing VAR-based and discriminative methods and propose a framework named Seg-VAR with autoregressive modelling that reconsiders segmentation as a conditional mask generation problem.
- We develop two critical strategies: Spatial-aware seglat encoding and image-seglat joint training. These designs enable our Seg-VAR to be adaptable for three segmentation settings.
- We conduct extensive experimental evaluations on challenging image segmentation benchmarks, including COCO, Cityscapes, and ADE20K, and the achieved state-of-the-art results demonstrate the effectiveness and generality of the proposed approach and shed new light on the autoregressive modeling segmentation strategy.

## 2   Related Work

**Image segmentation models.** Since the inception of FCN [46], semantic segmentation have flourished by various deep neural networks with the ability to classify each pixel. The follow-up methods then change focus to improve the limited receptive field of these models. PSPNet [78] and DeepLabV2 [7] aggregate multi-scale context between convolution layers. Sequentially, Nonlo-

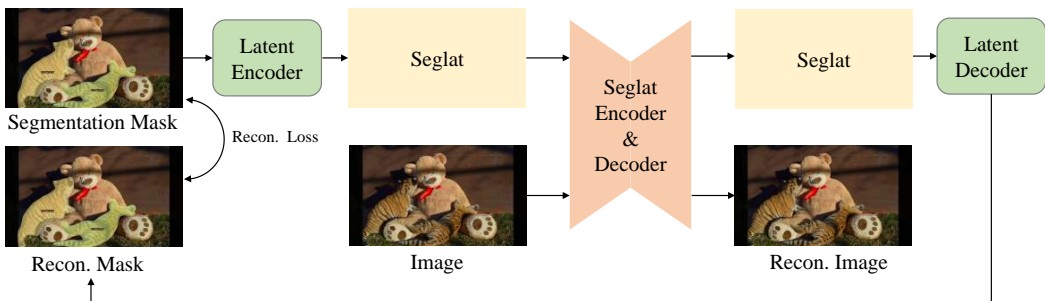

Figure 2: **Illustration of the latent and seglat learning** $(q_\phi, p_\theta)$ **of proposed Seg-VAR.** We first jointly model the seglat and image during training in the seglat encoder and decoder (red module). Then, with the well-trained encoder and decoder, we try to optimize the latent encoder and decoder (green module). Worth mentioning, we use different color in the binary segmentation mask to highlight different instances of the image.

cal [66], CCNet [30], and DGMN [76] integrate the attention mechanism in the convolution structure. Later on, Transformer-based methods (*e.g.* SETR [79] and Segformer [72]) are proposed following the introduction of Vision Transformers. More recently, MaskFormer [15] and Mask2Former [14] realize semantic segmentation with bipartite matching.

Specialized instance segmentation architectures are typically based upon "mask classification." They predict a set of binary masks each associated with a single class label. The pioneering work, Mask R-CNN [27], generates masks from detected bounding boxes. Follow-up methods either focus on detecting more precise bounding boxes [3, 6], or finding new ways to generate a dynamic number of masks, *e.g.*, using dynamic kernels [60, 67, 1] or clustering algorithms [35, 12]. Although the performance has been advanced in each task, these specialized innovations lack the flexibility to generalize from one to the other, leading to duplicated research effort. For instance, although multiple approaches have been proposed for building feature pyramid representations [43], as we show in our experiments, BiFPN [55] performs better for instance segmentation while FaPN [29] performs better for semantic segmentation.

Panoptic segmentation has been proposed to unify both semantic and instance segmentation tasks [34]. Architectures for panoptic segmentation either combine the best of specialized semantic and instance segmentation architectures into a single framework [73, 33, 12, 41] or design novel objectives that equally treat semantic regions and instance objects [4, 65]. Despite those new architectures, researchers continue to develop specialized architectures for different image segmentation tasks [53, 24]. We find panoptic architectures usually only report performance on a single panoptic segmentation task [65], which does not guarantee good performance on other tasks. For example, panoptic segmentation does not measure architectures' abilities to rank predictions as instance segmentations. Instead, here, we evaluate our Seg-VAR on all studied tasks to guarantee generalizability. Commonly, all the methods adopt the discriminative pixel-wise classification learning paradigm. This is in contrast to our generative image segmentation.

**Autoregressive models.** Autoregressive models, leveraging the powerful scaling capabilities of LLMs [51, 2, 17, 61, 62], use discrete image tokenizers [63, 52, 21] in conjunction with transformers to generate images based on next-token prediction. VQ-based methods [63, 52, 21, 38, 54] employ vector quantization to convert image patches into index-wise tokens and use a decoder-only transformer to predict the next token index. However, these methods are limited by the lack of scaled-up transformers and the quantization error inherent in VQ-VAE [63], preventing them from achieving performance on par with diffusion models. Parti [74], Emu3 [68], chameleon [57], loong [69] and VideoPoet [37] scaled up autoregressive models in text-to-image or video synthesis. Inspired by the global structure of visual information, Visual AutoRegressive modeling (VAR) redefines the autoregressive modeling on images as a next-scale prediction framework, significantly improving generation quality and sampling speed. HART [56] adopted hybrid tokenizers based on VAR. Fluid [23] proposed random-order models and employed a continuous tokenizer rather than a discrete tokenizer.

**Generative models for visual perception.** Image-to-image translation made one of the earliest attempts in generative segmentation, with far less success in performance [31]. Some good results

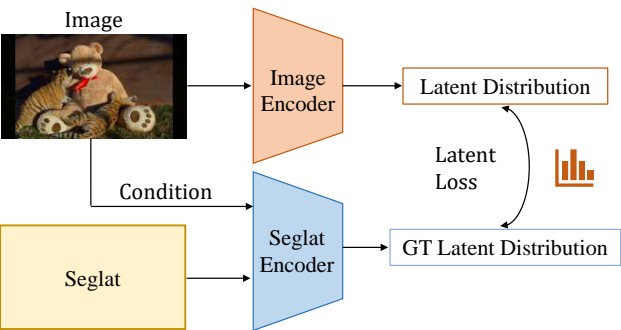

Figure 3: **Illustration of the latent learning ($p_\psi$) of proposed Seg-VAR.** In order to learn the latent representations, Seg-VAR optimizes the image encoder while freezing the seglat encoder. We also introduce the latent loss to minimize the divergence between two latent distributions for inference.

were achieved in limited scenarios such as face parts segmentation and Chest X-ray segmentation [39]. Replacing the discriminative classifier with a generative Gaussian Mixture model, GMMSeg [42] is claimed as generative segmentation, but the most is still of discriminative modeling. The promising performance of Pix2Seq [10] on several vision tasks leads to the prevalence of sequence-to-sequence task-agnostic vision frameworks. For example, Unified-I/O [48] supports a variety of vision tasks within a single model by seqentializing each task to sentences. Pix2Seq-D [9] deploys a hierarchical VAE (*i.e.* diffusion model) to generate panoptic segmentation masks. This method is inefficient due to the need for iterative denoising. UViM [36] realizes its generative panoptic segmentation by introducing latent variable conditioned on input images. It is also computationally heavy due to the need for model training from scratch. To address these issues, GSS introduces a notion of maskige for expressing segmentation masks in the form of RGB images, enabling the adoption of off-the-shelf data representation models (*e.g.* VGVAE) already pretrained on vast diverse imagery. However, the transformation of maskige is a simple MLP which restricts GSS from identifying the specific instances. Thus, we propose Seg-VAR with location-aware designs and hierarchical autoregressive modeling that solves the dilemma.

## 3  Method

### 3.1  Preliminaries

Conventionally, image segmentation can be formulated as a discriminative learning problem depending on the form of tasks:

$$
\begin{cases}
\max_{\pi} \log p_\pi(c \mid x), & \text{(Semantic Segmentation)} \\
\max_{\theta} \log p_\theta(c, y \mid x), & \text{(Instance Segmentation)} \\
\max_{\phi} \log p_\phi(c_{\text{stuff}}, c_{\text{things}}, y \mid x). & \text{(Panoptic Segmentation)}
\end{cases}
\tag{1}
$$

where $x \in \mathbb{R}^{H \times W \times 3}$ is an input image, $c \in \{0, 1\}^{H \times W \times K}$ is a *segmentation mask* in $K$ semantic categories, $y \in \mathbb{Z}^N$ is the instance number identifier, and $p_\pi, p_\theta, p_\phi$ are the discriminative pixel classifiers. Focusing on learning the classification boundary of input pixels, this approach enjoys high data and training efficiency [49].

In this work, we based our Seg-VAR on GSS [5] by introducing a discrete $L$-dimension latent distribution $q_\phi(z|c)$ (with $z \in \mathbb{Z}^L$) to the above log-likelihood as:

$$
\log p(c|x) \geq \mathbb{E}_{q_\phi(z|c)}\left[\log \frac{p(z, c|x)}{q_\phi(z|c)}\right],
$$

which is known as the Evidence Lower Bound (ELBO) [32], and we can easily expand it to instance and panoptic segmentation settings due to the chain rule and the independence of these variables (details are given in the supplementary material). And the ELBO can be written in the form of:

$$
\mathbb{E}_{q_\phi(z|c)}[\log p_\theta(c|z)] - D_{KL}\Big(q_\phi(z|c), p_\psi(z|x)\Big),
\tag{2}
$$

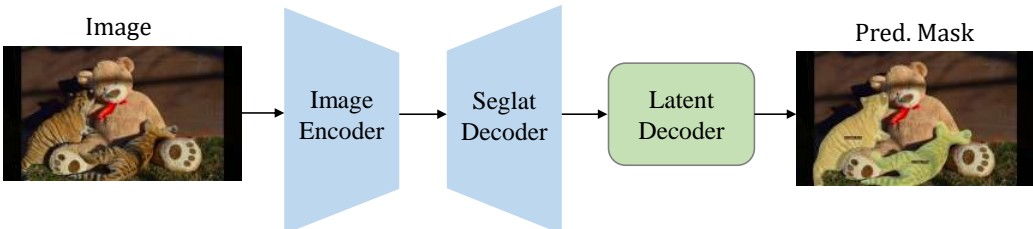

Figure 4: **Illustration of the inference stage.** The latent distribution generated by the image encoder is fed to the seglat decoder to generate the predicted seglat, and then finally generates the final prediction.

where we have three components in our formulation: (1) $p_\psi$: An **image encoder** (denoted as $\mathcal{I}_\psi$) that generates the prior distribution of latent tokens $z$ conditioned on the input image $x$. (2) $q_\phi$: A representative encoding function that encodes the semantic segmentation mask $c$ into discrete latent tokens $z$, which includes a seglat encoder (denoted as $\mathcal{E}_\phi$, implemented by VAR [64]) and a latent encoder ($\mathcal{T}_\phi$) that is built up with attention modules). (3) $p_\theta$: A function that decodes the semantic segmentation mask $c$ from the discrete latent tokens $z$, which includes a seglat decoder (denoted $\mathcal{D}_\theta$, implemented by VAR decoder [64]) and a latent decoder ($(\mathcal{T}_\theta)$).

## 3.2 Overall Architecture

As shown in Fig. 2, Seg-VAR mainly contains several modules: image encoder ($\mathcal{I}_\psi$), seglat encoder and decoder ($\mathcal{E}_\phi, \mathcal{D}_\theta$), and latent encoder as well as decoder ($\mathcal{T}_\phi, \mathcal{T}_\theta$).

**Image Encoder.** $\mathcal{I}_\psi$ is comprised of an image backbone (ResNet [28], Swin Transformer [45], etc.) and a Multi-scale fusion module. Multi-scale fusion is implemented with transformer layers and a projection layer. The output of the $\mathcal{I}_\psi$ is the latent token $z \in \mathbb{Z}^{H/d \times W/d}$.

**Latent Encoder and Decoder.** Latent encoder $\mathcal{T}_\phi$ on the other hand, is responsible for transforming the segmentation masks $\mathcal{M} \in \mathbb{R}^{H \times W \times N}$ into corresponding seglats $\mathcal{S} \in \mathbb{R}^{H \times W \times 3}$. We use transformer layers to generate the desired seglats. Thus, seglats can be viewed as a kind of RGB image. What's more, in order to be spatially-sensitive, we implement a colormap encoder $\Psi$ that converts the binary segmentation mask $M \in \{0, 1\}^{H \times W \times N}$ into an additional colormap $M_c \in \mathbb{R}^{H \times W \times 3}$ as:

$$M_c = \Psi(M), \tag{3}$$

where $N$ denotes the number of instances. $M_c$ is initialized to a zero value and then assigned to the corresponding color for each instance area by the spatial-aware color mapping. Inspired by UniGS [50], an image is partitioned into $a \times a$ grids, where each grid has an uniquely-assigned color. Each instance area is associated with these fixed colors if their gravity centers fall in the grids. To better distinguish the color difference, we select 6 candidate values {0, 51, 102, 153, 204, 255} for each RGB channel (if the categories are less than 124, then {0,64,128,192,255} is more preferred). Thus, the overall color number is $215 = 6^3 - 1$ (color (0,0,0) indicates the background). And the grid number $a^2$ should be less than 215. This location-aware color mapping can be effective because transformer design has position encoding that can help predict the corresponding colors. On the other hand, hand-crafted random assigning color will struggle to distinguish instances because of the large color space. Then, we concat the $\mathcal{S}$ with $M_c$, and the output is the final seglat feed into the seglat encoder. Unlike standard VAR, seglat tokens incorporate spatial information via a location-sensitive color mapping. This mapping assigns unique RGB values to instances based on their centroids (gridded into a*a regions), enabling the transformer to distinguish overlapping objects through positional cues.

**Seglat Encoder and Decoder.** Here, we adapt the design of ControlVAR (our design shares similarities with VQ-VAE but is specialized for segmentation, where we add extra image controls into modeling), and jointly model the image and seglat in each stage of the transformer structure. Let $k \in \{1, 2, \ldots, K\}$ denote a scale in the hierarchical transformer structure (where $S$ is the total number of scales). At scale $k$, image tokens and seglat tokens are tokenized by a shared tokenizer $\Phi$: Image features are $X_k \in [V]^{h_k \cdot w_k}$, and Seglats are $S_k \in [V]^{h_k \cdot w_k}$, where $V$ indicates the vocabulary size. A flatten operation is adopted to convert the sequence of 2D features into 1D. Full attention is

enabled for both control and image tokens belonging to the same scale, which allows the model to maintain spatial locality and to exploit the global context between control and image:

$$X'_k, S'_k = \text{Attention}(X_k, S_k, S_k). \tag{4}$$

Full attention is adopted for both seglat and image tokens within the same scale, so that the Seg-VAR can maintain the spatial locality and excavate the global connection within seglat and the input image. What's more, we employ the [CLS] and [TYP] as two pre-defined special tokens as start tokens. These two tokens are required to distinguish between different segmentation tasks. [CLS] provide semantic context for the generated image (categories), and [TYP] token is used to select the type of segmentation tasks. Both tokens are vital to Seg-VAR and can not be ablated. A standard cross entropy loss is utilized to supervise the seglat encoder and decoder in the reconstruction process.

### 3.3 Multi-stage Training

Our training can be divided into several stages: 1) the learning of the seglat encoder and decoder ($\mathcal{E}_\phi, \mathcal{D}_\theta$). 2) the learning of the latent encoder and decoder ($\mathcal{T}_\phi, \mathcal{T}_\theta$). 3) the image encoder ($\mathcal{I}_\psi$). Namely, the first and second stage are illustrated in Fig. 2 (which is the posterior modules of latent tokens), while the third stage is shown in Fig. 3 (which is the prior module of latent tokens). We now delve into details of the multiple stages of posterior and prior training process.

**Stage 1: Seglat Learning.** In the first stage is mainly about the joint training of image with seglat, which is discussed in ControlVAR, except that we novelly change the control signal to a unique type of RGB image, seglat.

**Stage 2: Latent Learning.** In the second stage, the training is mainly to optimize:

$$\min_{\theta,\phi} \mathbb{E}_{q_\phi(z|c)} \|p_\theta(c|z) - c\|. \tag{5}$$

However, we introduce the latent encoder and decoder to transform the $c$, and now we have:

$$\min_{\hat{\phi},\hat{\theta}} \mathbb{E}_{q_{\hat{\phi}}(\hat{z}|\mathcal{T}_\phi(c))} \|\mathcal{D}_{\hat{\theta}}(\hat{z}) - \mathcal{T}_\phi(c)\| + \min_{\mathcal{T}_\theta} \mathbb{E}_{q_{\hat{\phi}}(\hat{z}|\mathcal{T}_\phi(c))} \|\mathcal{T}_\theta(\mathcal{D}_{\hat{\theta}}(\hat{z})) - c\|,$$

where $\mathcal{T}_\phi(c) = \mathcal{S}$, which is the seglat (we consider as the latent representation of segmentation mask, and name this for convenience). The first term refers to image reconstruction, while the second term can be simplified as:

$$\min_{\mathcal{T}_\theta} \mathbb{E}_{q_{\hat{\phi}}(\hat{z}|\mathcal{T}_\phi(c))} \|\mathcal{T}_\theta(\mathcal{S}) - c\|, \tag{6}$$

where the training parameters are far less than training VAR from scratch.

**Stage 3: Image Encoder Learning.** As shown in Fig. 3, the optimization target of this stage should be to minimize the distance between the distribution predicted by $\mathcal{I}_\psi$ and the latent codes generated by seglat encoder (VAR). We use the cross-entropy loss to measure the distance of this distance:

$$\min_\psi D_{KL}\Big(q_\phi(z|c), p_\psi(z|x)\Big). \tag{7}$$

### 3.4 Inference

As illustrated in Fig. 4, during inference, we first take the latent tokens $z$ that are predicted by the image encoder $\mathcal{I}_\psi$, and feed them into the latent decoder $\mathcal{D}_\theta$ to generate the predicted latent concatenation $\hat{x}^{(c)}$. Next, we apply the inverse transform decoding to the predicted latents to obtain the final segmentation mask $\hat{c}$.

## 4 Experiments

In the first part, we present the evaluation datasets and metrics. Then we present the training settings. Finally, we demonstrate that Seg-VAR is an effective generative architecture for universal image segmentation through comparisons with different specialized methods on standard benchmarks. We evaluate our model on all three tasks, which all obtain the state-of-the-art results.

**Datasets.** We study Seg-VAR using four widely used image segmentation datasets that support semantic, instance and panoptic segmentation: COCO [44] (80 "things" and 53 "stuff" categories),

| Method | Backbone | PQ | $PQ^{Th}$ | $PQ^{St}$ | $AP^{Th}_{pan}$ | $mIoU_{pan}$ | #params. | FLOPs | fps |
|---|---|---|---|---|---|---|---|---|---|
| DETR [4] | R50 | 43.4 | 48.2 | 36.3 | 31.1 | - | - | - | - |
| MaskFormer [16] | R50 | 46.5 | 51.0 | 39.8 | 33.0 | 57.8 | 45M | 181G | 17.6 |
| Mask2Former [14] | R50 | 51.9 | 57.7 | 43.0 | 41.7 | 61.7 | 44M | 226G | 8.6 |
| Seg-VAR | R50 | **54.1** | **60.1** | **45.8** | **44.3** | **64.2** | 315M | 605G | 5.2 |
| DETR [4] | R101 | 45.1 | 50.5 | 37.0 | 33.0 | - | - | - | - |
| MaskFormer [16] | R101 | 47.6 | 52.5 | 40.3 | 34.1 | 59.3 | 64M | 248G | 14.0 |
| Mask2Former [14] | R101 | 52.6 | 58.5 | 43.7 | 42.6 | 62.4 | 63M | 293G | 7.2 |
| **Seg-VAR** | R101 | **54.7** | **60.4** | **46.2** | **44.5** | **64.6** | 335M | 624G | 4.6 |
| Max-DeepLab [65] | Max-L | 51.1 | 57.0 | 42.2 | - | - | 451M | 3692G | - |
| MaskFormer [16] | Swin-L[†] | 52.7 | 58.5 | 44.0 | 40.1 | 64.8 | 212M | 792G | 5.2 |
| K-Net [77] | Swin-L[†] | 54.6 | 60.2 | 46.0 | - | - | - | - | - |
| Mask2Former [14] | Swin-L[†] | 57.8 | 64.2 | 48.1 | 48.6 | 67.4 | 216M | 868G | 4.0 |
| GSS [5] | Swin-L[†] | 44.9 | 50.2 | 32.6 | 36.9 | 54.2 | 386M | 1142G | 3.4 |
| **Seg-VAR** | Swin-L[†] | **59.7** | **65.6** | **50.5** | **49.6** | **68.7** | 522M | 1320G | 3.2 |

Table 1: **Panoptic segmentation on COCO panoptic** `val2017` **with 133 categories.** Seg-VAR consistently outperforms Mask2Former [14] by a large margin on all metrics. Our best model outperforms prior state-of-the-art models by 1.9 PQ and GSS [77] by 14.8 PQ. Backbones pre-trained on ImageNet-22K are marked with [†].

| Method | Backbone | AP | $AP^{S}$ | $AP^{M}$ | $AP^{L}$ | $AP^{boundary}$ | #params. | FLOPs | fps |
|---|---|---|---|---|---|---|---|---|---|
| MaskFormer [16] | R50 | 34.0 | 16.4 | 37.8 | 54.2 | 23.0 | 45M | 181G | 19.2 |
| Mask R-CNN [27] | R50 | 37.2 | 18.6 | 39.5 | 53.3 | 23.1 | 44M | 201G | 15.2 |
| Mask R-CNN [27] | R50 | 42.5 | 23.8 | 45.0 | 60.0 | 28.0 | 46M | 358G | 10.3 |
| Mask2Former [11] | R50 | 43.7 | 23.4 | 47.2 | 64.8 | 30.6 | 44M | 226G | 9.7 |
| **Seg-VAR** | R50 | **45.8** | **25.2** | **49.8** | **68.1** | **33.4** | 315M | 605G | 5.9 |
| Mask R-CNN [27] | R101 | 38.6 | 19.5 | 41.3 | 55.3 | 24.5 | 63M | 266G | 10.8 |
| Mask R-CNN [27] | R101 | 43.7 | 24.6 | 46.4 | 61.8 | 29.1 | 65M | 423G | 8.6 |
| Mask2Former [11] | R101 | 44.2 | 23.8 | 47.7 | 66.7 | 31.1 | 65M | 293G | 7.8 |
| **Seg-VAR** | R101 | **46.5** | **25.2** | **49.6** | **70.1** | **34.6** | 335M | 624G | 5.2 |
| QueryInst [24] | Swin-L[†] | 48.9 | 30.8 | 52.6 | 68.3 | 33.5 | - | - | 3.3 |
| Swin-HTC++ [6] | Swin-L[†] | 49.5 | 31.0 | 52.4 | 67.2 | 34.1 | 284M | 1470G | - |
| Mask2Former | Swin-L[†] | 50.1 | 29.9 | 53.9 | 72.1 | 36.2 | 216M | 868G | 4.0 |
| **Seg-VAR** | Swin-L[†] | **52.7** | **31.2** | **55.2** | **75.4** | **39.4** | 522M | 1320G | 3.2 |

Table 2: **Instance segmentation on COCO** `val2017` **with 80 categories.** Seg-VAR outperforms strong Mask2Former [14] baselines for both AP and $AP^{boundary}$ [13] metrics. Our Seg-VAR surpasses Mask2Former by a huge 2.6 AP on the largest backbone Swin-L, and demonstrates superior performance across all metrics and backbones. Backbones pre-trained on ImageNet-22K are marked with [†].

ADE20K [80] (100 "things" and 50 "stuff" categories), and Cityscapes [18] (8 "things" and 11 "stuff" categories). Panoptic and semantic segmentation tasks are evaluated on the union of "things" and "stuff" categories, while instance segmentation is only evaluated on the "things" categories.

**Evaluation metrics.** For *panoptic segmentation*, we use the standard **PQ** (panoptic quality) metric [34]. We further report $\mathbf{AP^{Th}_{pan}}$, which is the AP evaluated on the "thing" categories using instance segmentation annotations, and $\mathbf{mIoU_{pan}}$, which is the mIoU for semantic segmentation by merging instance masks from the same category, of the same model trained *only* with panoptic segmentation annotations. For *instance segmentation*, we use the standard **AP** (average precision) metric [44]. For *semantic segmentation*, we use **mIoU** (mean Intersection-over-Union) [22].

| Method | Pretrain | Backbone | Iteration | mIoU |
|---|---|---|---|---|
| *- Discriminative modeling:* | | | | |
| FCN [46] | 1K | ResNet-101 | 80k | 77.02 |
| PSPNet [78] | 1K | ResNet-101 | 80k | 79.77 |
| DeepLab-v3+ [8] | 1K | ResNet-101 | 80k | 80.65 |
| NonLocal [66] | 1K | ResNet-101 | 80k | 79.40 |
| CCNet [30] | 1K | ResNet-101 | 80k | 79.45 |
| Maskformer [15] | 1K | ResNet-101 | 90k | 78.50 |
| Mask2former [14] | 1K | ResNet-101 | 90k | 80.10 |
| SETR [79] | 22K | ViT-Large | 80k | 78.10 |
| UperNet [71] | 22K | Swin-Large | 80k | 82.89 |
| Mask2former [14] | 22K | Swin-Large | 90k | **83.30** |
| SegFormer [72] | 1K | MiT-B5 | 160k | 82.25 |
| *- Generative modeling:* | | | | |
| UViM† [36] | 22K | Swin-Large | 160k | 70.77 |
| GSS-FF [5] | 22K | Swin-Large | 80k | 78.90 |
| GSS-FT-W [5] | 22K | Swin-Large | 80k | 80.05 |
| Seg-VAR | 22K | Swin-Large | 80k | **85.82** |

Table 3: **Semantic Segmentation on the Cityscapes** `val` **split:** UViM† [36] is reproduced on PyTorch. "1K" means pretrained on ImageNet 1K [19] while "22K" means pretrained on ImageNet 22K [19]. Our model surpasses previous state-of-the-art by 2.52 mIoU, demonstrating the effectiveness of Seg-VAR.

| Method | Pretrain | Backbone | Iteration | mIoU |
|---|---|---|---|---|
| *- Discriminative modeling:* | | | | |
| FCN [46] | 1K | ResNet-101 | 160k | 41.40 |
| CCNet [30] | 1K | ResNet-101 | 160k | 43.71 |
| DANet [25] | 1K | ResNet-101 | 160k | 44.17 |
| UperNet [71] | 1K | ResNet-101 | 160k | 43.82 |
| Deeplab-v3+ [8] | 1K | ResNet-101 | 160k | 45.47 |
| Maskformer [15] | 1K | ResNet-101 | 160k | 45.50 |
| Mask2former [14] | 1K | ResNet-101 | 160k | 47.80 |
| OCRNet [75] | 1K | HRNet-W48 | 160k | 43.25 |
| SegFormer [72] | 1K | MiT-B5 | 160k | **50.08** |
| SETR [79] | 22K | ViT-Large | 160k | 48.28 |
| *- Generative modeling:* | | | | |
| UViM† [36] | 22k | Swin-Large | 160k | 43.71 |
| GSS-FF [5] | 22K | Swin-Large | 160k | 46.29 |
| GSS-FT-W [5] | 22K | Swin-Large | 160k | 48.54 |
| Seg-VAR | 22K | Swin-Large | 160k | **54.90** |

Table 4: **Semantic Segmentation comparison with previous art methods on the ADE20K** `val` **split**. "1K" means pretrained on ImageNet 1K [19] while "22K" means pretrained on ImageNet 22K [19]. Our model surpasses previous state-of-the-art by 4.82 mIoU, demonstrating the effectiveness of Seg-VAR.

## 4.1 Training settings

**Panoptic and instance segmentation.** We operate all experiments with 8 V100 GPUs. We use Detectron2 [70] and follow the updated Mask R-CNN [27] baseline settings for the COCO dataset. More specifically, we use AdamW [47] optimizer and the step learning rate schedule. We use an initial learning rate of $0.0001$ and a weight decay of $0.05$ for all backbones. A learning rate multiplier of $0.1$ is applied to the backbone and we decay the learning rate at $0.9$ and $0.95$ fractions of the total number of training steps by a factor of 10. Training iterations are also reported in all experimental figures. For data augmentation, we use the large-scale jittering (LSJ) augmentation [26, 20] with a random scale sampled from the range 0.1 to 2.0 followed by a fixed size crop to $1024 \times 1024$. We use the standard Mask R-CNN inference setting where we resize an image with shorter side to 800 and longer side up-to 1333. We also report FLOPs and fps. FLOPs are averaged over 100 validation images (COCO images have varying sizes). Frames-per-second (fps) is measured on a V100 GPU with a batch size of 1 by taking the average runtime on the entire validation set including post-processing time.

**Semantic segmentation.** We follow the same settings as [11] to train our models, except: 1) a learning rate multiplier of 0.1 is applied to *both* CNN and Transformer backbones instead of only applying it to CNN backbones in [16], 2) both ResNet and Swin backbones use an initial learning rate of 0.0001 and a weight decay of 0.05, instead of using different learning rates in [16].

**VAR modeling.** We follow VAR [59] and ControlVAR [40]. During training, we leverage the pre-trained VAR tokenizer to tokenize seglat and control. The training details follow the strategy in ControlVAR, which refers to an approach of sampling both pixel- and token-level controls for image generation with teacher-forcing guidance. For inference, we utilize top-k top-p sampling with k=900 and p=0.96 for encoding and decoding the seglat. As for the training objectives, the training objective is based on the Evidence Lower Bound (ELBO), optimizing three components: (1) mask reconstruction loss via seglat decoder, (2) KL divergence to align image-derived latents with seglat distributions, and (3) cross-entropy loss for token prediction.

## 4.2 Main results

**Panoptic segmentation.** In Table. 1, we compare Seg-VAR with state-of-the-art models for panoptic segmentation on the COCO panoptic [34] dataset validation split. Seg-VAR consistently outperforms Mask2Former by 1.9. With Swin-L backbone, our Seg-VAR sets a new state-of-the-art of 59.7

| ID | Seglat Learning Stage 1 | Img. Enc. Learning Stage 3 | ADE20K mIoU | COCO AP |
|----|-------------------------|----------------------------|-------------|---------|
| 1 |   |   | 78.9 | 46.2 |
| 2 | ✓ |   | 83.4 | 52.0 |
| 3 |   | ✓ | 81.6 | 49.3 |
| 4 | ✓ | ✓ | **85.8** | **52.7** |

Table 5: **Ablation on the key design of Seg-VAR.** These results demonstrate the effectiveness of our designs and training strategy.

| Generation Model | ADE20K | COCO |
|------------------|--------|------|
| VQGAN | 74.6 | 42.8 |
| DALL·E 2 | 80.2 | 47.9 |
| SD-XL | 81.8 | 48.9 |
| VAR | **85.8** | **52.7** |

Table 6: **Ablation on different generation models**. We experimented on different image generation models, the results indicate the superiority of VAR in segmentation tasks.

PQ, outperforming existing state-of-the-art [14] by 1.9 PQ and generative method GSS by 14.8 PQ. This indicates the effectiveness of our jointly modeling strategy with specially-designed generative encoders and decoders, which successfully encode localization information as well as instance information. GSS, on the other hand, fails to identify different instances effectively.

Beyond the PQ metric, our Seg-VAR also achieves higher performance on two other metrics compared to Mask2Former: $AP_{pan}^{Th}$, which is the AP evaluated on the 80 "thing" categories using *instance segmentation annotation*, and $mIoU_{pan}$, which is the mIoU evaluated on the 133 categories for semantic segmentation converted from panoptic segmentation annotation. This shows Seg-VAR's universality: Even trained *only* with panoptic segmentation annotations, it can be used for instance, and semantic segmentation.

**Instance segmentation.** We compare Seg-VAR with state-of-the-art models on the COCO [44] dataset in Table. 2. With the Swin-L backbone, Seg-VAR outperforms the state-of-the-art Mask2Former by 2.6 AP and 3.2 $AP^{boundary}$. On other backbones, including R50 and R101, our Seg-VAR still shows superiority over previous approaches across all metrics (2.3 AP and 2.1 AP, respectively). These results further validate the efficacy of our jointly hierarchical modeling strategy with location-aware generative segmentation latent encoders and decoders, which successfully encode localization information as well as instance information.

**Semantic segmentation.** We compare Seg-VAR with SOTA models for semantic segmentation on the Cityscapes [18] dataset in Table. 3. With the Swin-L backbone, Seg-VAR outperforms previous SOTA methods, including Mask2Former [14] with a 2.52 increase in fewer training iterations, and a huge boost of 5.77 mIoU compared to the previous generative segmentation model GSS.

We also compare Seg-VAR with state-of-the-art models for semantic segmentation on the ADE20K [80] dataset in Table. 4. Seg-VAR outperforms previous SOTA methods, including Mask2Former [14] with an increase of 7.1 mIoU and SegFormer with a 4.82 improvement. What's more, our Seg-VAR outperforms GSS with a 6.4 increase, which is a large margin. This should credit to the modeling and novel design in our latent encoders and decoders.

The consistent superiority of our framework across both datasets (Table. 3, 4) empirically validates its capacity to reconcile structural priors with discriminative feature learning. These results highlight the critical role of our architectural innovations, particularly the synergistic design of latent encoders for disentangled representation learning and decoders for geometry-aware refinement, in advancing semantic segmentation performance.

## 4.3 Ablation studies

In this part, we ablate various key designs of our Seg-VAR from different aspects, ranging from the ablation of key designs, choice of generation models, parameter efficiency, and hyper parameters of grid size and palette.

**Key design of Seg-VAR.** In Table. 5, we demonstrate the effectiveness of our key components design and corresponding multi-stage training strategy. stage 1 represents seglat encoder/decoder, stage 2 refers to latent encoder and decoder, and stage 3 represents image encoder learning. Since latent learning is core idea of Seg-VAR, we keep the latent encoder and decoder in ablation studies (in Table. 7). As shown in the table, with the seglat learning strategy implemented, the performance

| Method | Dataset | mIoU |
|--------|---------|------|
| Vanilla VAR | ADE20K | 77.4 |
| Seg-VAR | ADE20K | 85.8 |

| Method | Dataset | mIoU |
|--------|---------|------|
| Mask2Former | R50 | 63.8 |
| Seg-VAR | R50 | 64.2 |

| Grid Number | mIoU | Palette Size | mIoU |
|-------------|------|--------------|------|
| 4 | 84.4 | 124 | 85.4 |
| 8 | 85.2 | 215 | 85.8 |
| 12 | 85.8 | 342 | 85.3 |

Table 7: **Ablation on the key design of Seg-VAR.** These results demonstrate that simply using vanilla VAR without seglat modules, the result is 8.4 lower than our Seg-VAR.

Table 8: **Ablation on the parameters.** These results demonstrate that under comparable parameters, our Seg-VAR still outperforms Mask2Former in the R50 backbone.

Table 9: **Ablation on grid number and palette size**. We experimented on different settings of grid number and palette size, the results indicate the robustness of these variations.

improves greatly, showing 4.5 mIoU and 5.8 AP enhancement. The image encoder learning strategy also shows great effectiveness with a 2.7 and 3.1 mIoU improvement, respectively. By implementing these designs, our model is capable of harmonize semantic accuracy with geometric consistency.

**Different generation model designs.** In Table. 6, we show that VAR is better than previous VQ-VAE and diffusion-based generation models. With VAR as the encoder, our model surpasses SD-XL by 4.0 mIoU and 3.8 AP. This indicate that VAR is capable of being implemented in general image segmentation tasks for its superior structure of autoregressive modeling.

**Seglats designs.** To explicitly validate the role of seglats, we conducted additional experiments using a "vanilla VAR" baseline (remove latent encoder and decoder, and using plain VAR). As shown in the Table. 7, simply using vanilla VAR without seglat modules and tested on ADE2OK, the result is 8.4 lower than our Seg-VAR. This indicate the importance of latent learning strategy in our Seg-VAR.

**Model parameters.** In Table. 8, we examined the model parameter efficiency. To better compare our model with traditional discriminative segmentation models, we adjust the parameters of Mask2Former by extending its transformer layer number so that the parameters can be comparable. As shown in the table, our Seg-VAR still outperforms Mask2Former by 0.4 mIoU in COCO panoptic dataset.

**Sensitivity of grid size and colormap.** We evaluate the robustness of grid number and Palette on ADE20K. As shown in the Table. 9, the performance decreases as the number decreases, because the granularity can help model better distinguish instances. As for the palette size of colormap, we discover that 6 is the optimal number because we have to balance between a large color space and loss of generality. (Ideally, the size should be larger than the category). We find that grid numbers are more sensitive than palette size, but they still contribute to the performance gain. These results confirm that Seg-VAR is robust to reasonable variations in grid/color size.

## 5 Conclusions

In this work, we analyze the limitations of existing VAR-based and discriminative methods and propose a framework named Seg-VAR with autoregressive modelling that reconsiders segmentation as a conditional mask generation problem. We develop two critical strategies: Spatial-aware seglat encoding and image-seglat joint training. These designs enable our Seg-VAR to be adaptable for three segmentation settings. By decomposing segmentation into a coarse-to-fine token prediction process, Seg-VAR bridges the gap between autoregressive modeling's sequential dependency learning and segmentation's demand for precise spatial reasoning. Our experiments demonstrate that autoregressive methods, long dominant in generation tasks, can rival and even surpass parallel architectures in segmentation accuracy.

**Limitations and Broader Impact.** Even though our model demonstrates great potential in generating high-quality segmentation masks, its application to video domains is yet to be discovered. Also, due to the memory of image generation models, the memory cost is larger than transformer-based segmentation models. As for broader impact, we believe our work lay a foundation for future works in unifying generation and perception tasks.

**Acknowledgement.** This work is supported by the National Natural Science Foundation of China (No. 62422606, 62201484, 624B2124) and the computation resources provided by Shanghai Artificial Intelligence Laboratory.

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
