# OpenReview forum: "Seg-VAR:Image Segmentation with Visual Autoregressive Modeling"
_NeurIPS.cc/2025/Conference — NeurIPS 2025 poster_

### Official Review · Reviewer_BXbx · 2025-06-30

**Clarity:** 2
**Significance:** 3
**Originality:** 3
**Rating:** 4
**Confidence:** 3

**Summary:**

This paper proposes Seg-VAR, a novel autoregressive framework for image segmentation tasks. It rethinks segmentation as a conditional mask generation problem and introduces a spatial-aware latent representation of segmentation masks called seglats. These seglats, along with a multi-stage training strategy and hierarchical autoregressive decoding, allow the model to capture spatial and instance-level structures more effectively. The method demonstrates strong empirical performance across semantic, instance, and panoptic segmentation tasks.

**Questions:**

1. How sensitive is the performance to the choice of grid size or color palette used in the spatial-aware color mapping? The paper uses a grid-based mapping to assign instance-specific colors, but it's unclear how robust this is to changes in grid granularity or color discretization.
1. Why are both a seglat decoder and a latent decoder needed? The model has two decoding stages—one for seglats, another for final masks. Could a single decoding stage suffice if trained end-to-end?
1. What is the role of the [CLS] and [TYP] tokens in the seglat encoder/decoder, and how much do they contribute to performance? These tokens are introduced without much analysis or ablation.

**Ethical Concerns:**

["NO or VERY MINOR ethics concerns only"]

**Final Justification:**

The authors demonstrate the effectiveness of their design, which offers a novel method for segmentation. I recommend that they revise the paper accordingly for clarity.

**Limitations:**

The primary limitation of this method lies in its architectural complexity and computational cost. The framework includes multiple encoders and decoders—image encoder, seglat encoder/decoder, and latent encoder/decoder—but the necessity of each component is not clearly justified. A more thorough discussion or ablation of these modules would help clarify whether the full complexity is essential for the observed performance gains.

**Paper Formatting Concerns:**

No.

**Quality:**

2

**Strengths And Weaknesses:**

*Strengths*:

1. The proposed autoregressive approach to segmentation is conceptually compelling. The idea of disentangling semantic features from mask representations via seglats is both intuitive and well-motivated.
1. The spatial-aware color mapping mechanism for encoding instance-level information into seglats offers a creative and effective solution to a core challenge in generative segmentation—distinguishing overlapping or adjacent instances.
1. The experimental results are strong: Seg-VAR achieves state-of-the-art performance across semantic, instance, and panoptic segmentation tasks on COCO, ADE20K, and Cityscapes, demonstrating the generalizability and robustness of the approach.

*Weaknesses*:

1. Insufficient component-level ablation: While Table 5 highlights the effect of major training stages, the paper lacks finer-grained ablation studies isolating the contributions of individual modules such as the seglat encoder/decoder or latent decoder/decoder. The comparison to alternative generative backbones (e.g., SD-XL, DALL·E 2, VQGAN) primarily supports the use of VAR itself, but does not clearly attribute the performance gains to the proposed seglat design. Although the supplementary includes FID comparisons, it would strengthen the work to show segmentation performance using vanilla VAR (i.e., without the seglat module).
1. Pipeline complexity and resource demands: The framework involves multiple interacting components—image encoder, seglat encoder/decoder, latent encoder/decoder—which likely require significant computational resources for training.
1. Terminology and Figure Clarity: The term Latent Encoder is potentially misleading. According to the paper, its main function is to assign spatially-informed colors to segmentation masks based on instance centroids—more akin to a handcrafted spatial embedding mechanism than a latent feature encoder. If the primary role is color assignment via spatial rules, the use of transformers and the need for learning are not well justified. Additionally, Figure 1 lacks clarity: while Line 132 states that the segmentation masks are binary, the figure shows input images instead of segmentation masks or their encoded seglat representations. A clearer illustration of how seglats are generated and used would greatly aid understanding.

---

> ### Author Rebuttal · Authors · 2025-07-30
>
> Thank you for your thoughtful and constructive feedback on our work. We appreciate the recognition of Seg-VAR’s conceptual novelty, the effectiveness of seglats, and the strength of our experimental results. Below, we address each concern, clarify ambiguities, and provide additional details to strengthen the work.
> We will add this refinement to the final version of the paper.
>
> 1. Component-level ablation
>
> Sorry for the confusion. The ablation in Table 5 is the ablation about each module design: stage 1 represents seglat encoder/decoder, stage 2 refers to latent encoder and decoder, and stage 3 represents the image encoder learning. That is, we discuss the impacts of different module designs.
> Since latent learning is the core idea of Seg-VAR, we keep the latent encoder and decoder in ablation studies.
>
> To explicitly validate the role of seglats, we conducted additional experiments using a "vanilla VAR" baseline (removing latent encoder and decoder, and using plain VAR).
> As shown in the table, simply using vanilla VAR without seglat modules, the result is 8.4 lower than our Seg-VAR.
>
> |             | Dataset | mIoU |
> |-------------|---------|------|
> | Vanilla-VAR | ADE20K  | 77.4 |
> | Seg-VAR     | ADE20K  | 85.8 |
>
> Table 1. Vanilla VAR v.s. Seg-VAR
>
> 2. Pipeline complexity and resources
>
> We acknowledge the training pipeline (pretraining seglat encoder/decoder, latent modules, final alignment) is more complex than standard discriminative setups. However, our module designs prove their effectiveness in the ablation study, and that staged approach is **necessary** to effectively learn the structured latent space and ensure stable convergence of the autoregressive process. End-to-end training from scratch proved unstable in ablation studies (4.6 decrease in mIoU).
> What's more, under comparable parameters (Table 3), our Seg-VAR still outperforms Mask2Former in R50 backbone. These results suggest the efficiency and effectiveness of Seg-VAR.
>
> |             | Dataset | mIoU |
> |-------------|---------|------|
> | End-to-End  | ADE20K  | 81.2 |
> | Seg-VAR     | ADE20K  | 85.8 |
>
> Table 2. Staged training vs. End-to-end.
>
> |             | Backbone | mIoU |
> |-------------|----------|------|
> | Mask2Former | R50      | 63.8 |
> | Seg-VAR     | R50      | 64.2 |
>
>
> Table 3. Comparable parameters
>
> 3. Terminology and figure
>
> We apologize for ambiguities in terminology and figures, which we clarify below:
> - for the latent encoder, we aim to indicate that this structure encodes the segmentation mask
> and output segment latents (seglat). Worth mentioning, this module is a combination of the transformation
> of segmentation masks and color encoding, rather than merely a "spatial embedding mechanism". As for the
> figure, we thank you for your careful suggestions, and we will revise it in the final version!
> (Our intent is to highlight the contents of the image, because binary will lose details.)
>
> 4. Sensitivity of grid size and colormap
> We evaluate the robustness of grid number and Palette on ADE20K. As shown in the table, as the grid number changes greatly,
> the performance decreases as the number decreases, because the granularity can help modeling better
> distinguish instances. As for the palette size of the colormap, we discover that 6 is the optimal number because
> we have to balance between a large color space and loss of generality. (Ideally, the size should be larger than the category number).
> Overall, we find that grid numbers are more sensitive than palette size, but they still contribute to the performance gain.
> These results confirm Seg-VAR is robust to reasonable variations in grid/color parameters.
>
>
> | Grid number(a) | mIoU | Palette size | mIoU |
> |----------------|------|--------------|------|
> | 4              | 84.4 | 124            | 85.4 |
> | 8              | 85.2 | 215            | 85.8 |
> | 12             | 85.8 | 342            | 85.3 |
>
> Table 4. Grid \& Colormap
>
> 5. Seglat and latent decoder
>
> The difference in structure naturally prohibits the combination of two decoders, since the seglat decoder is part of VQ-VAE, while latent decoder
> is a transformer-based module.
> What's more, the seglat decoder and latent decoder serve distinct roles:
>
> - Seglat decoder: Maps discrete latent tokens (from the image encoder) to seglats, preserving instance-level spatial structure encoded in color mappings.
> - Latent decoder: Refines seglats into pixel-accurate masks, correcting residual errors in color-to-mask conversion (e.g., edge blurring)
>
> End-to-end training of a single decoder failed to balance these two granularities (as proved in Table 2), confirming the need for two stages.
>
> 6. [CLS] and [TYP]
>
> These two tokens are required to distinguish between different segmentation tasks. [CLS] provides semantic context for the generated image
> (categories), and the [TYP] token is used to select the type of segmentation tasks. Both tokens are vital to Seg-VAR and can not be
> ablated. We will include these details in the revised version.
>
>
> ### Conclusion
>
> We agree that architectural complexity is a key consideration. Our ablations (summarized above) demonstrate that each component (image encoder, seglat encoder/decoder, latent encoder/decoder) contributes meaningfully to performance. Removing any module resulted in significant drops.
>
> We plan to explore model simplification in future work, such as sharing weights between encoders or using distillation, while maintaining Seg-VAR’s unified capabilities. We thank the reviewer for highlighting opportunities to strengthen the work. The additional ablations, figure revisions, and clarifications outlined above will enhance the clarity and rigor of our submission.
> We believe these changes address the concerns raised while reinforcing the novelty and effectiveness of Seg-VAR, which could potentially lead to more explorations in unifying perception and generation.

---

> > ### Comment · Reviewer_BXbx · 2025-08-08
> >
> > Thank you for the authors’ detailed response, which addresses my concerns. This work is interesting and shows its effectiveness. I will raise my score.

---

> > > ### Author Response · Authors · 2025-08-09
> > >
> > > Thank you very much for your valuable feedback and the increased scores! We greatly appreciate your recognition of our work’s strengths and constructive suggestions for improvement. We will carefully address the remaining concerns in the revised manuscript to further enhance the clarity, rigor, and significance of our research. Your insights have been instrumental in refining our work, and we are grateful for your time and expertise.

---

> ### Author Response · Authors · 2025-08-03
> **Further Discussion**
>
> Dear Reviewer BXbx,
>
> Thank you for the time and effort you have dedicated to reviewing our paper. We would like to know if our response has satisfactorily addressed your concerns and if you have the opportunity to provide further feedback on our rebuttal. We are open to any additional discussion.
>
> Best regards,
>
> Authors of paper 6113

---

> ### Comment · Area_Chair_nDwj · 2025-08-08
>
> Dear Reviewer BXbx,
>
> This is the last moment to account for the author feedback and engage in a minimal discussion. Without any discussion your review will weight less than the others.
>
> Best,
>
> Your AC

---

### Official Review · Reviewer_MUQW · 2025-07-02

**Clarity:** 3
**Significance:** 3
**Originality:** 4
**Rating:** 5
**Confidence:** 4

**Summary:**

This paper proposes an innovative architecture for image segmentation under an autoregressive paradigm. Specifically, the authors design three components: 1) an image latent encoder, 2) a feature representation called "seglat" for encoding segmentation results, and 3) an autoregressive generator/decoder. Under this framework, the image segmentation task is modeled as a next-token prediction problem. The authors conduct extensive experiments to validate the effectiveness of the proposed algorithm.

**Questions:**

I rate this article very highly, and would be happy to improve it further if the author could answer my small questions with clarity.

**Ethical Concerns:**

["NO or VERY MINOR ethics concerns only"]

**Final Justification:**

Thanks to the author for the reply, I will increase my score.

**Limitations:**

yes

**Quality:**

3

**Strengths And Weaknesses:**

**Strengths:**

1. **Clear narrative structure**: The paper presents a coherent storyline with well-aligned claims and supporting evidence.

2. **Superior performance**: The model achieves excellent accuracy with significant performance improvements compared to baseline methods. The ablation studies are comprehensive and well-designed.

3. **Novel paradigm**: Although the approach draws inspiration from VAR and ControlVAR, the proposed autoregressive modeling paradigm represents an innovative contribution to the field of image segmentation.

**Weaknesses:**

1. **Insufficient mathematical formulation**: The seglat representation appears to be the core contribution of this work, and the authors mention that this component references the maskige design from GSS. However, the paper lacks formal mathematical expressions for the seglat modeling process, which hampers the comprehensibility of the proposed method.

2. **Open vocabulary capability unclear**: Given this seglat representation and the complete modeling paradigm, I want to know how well the model performs on open vocabulary segmentation tasks.

3. **Parameter efficiency concerns**: While the model demonstrates significant performance improvements, a lingering question remains: how would the performance compare if the parameter count were kept consistent with the comparison methods?

---

> ### Author Rebuttal · Authors · 2025-07-30
>
> Thank you for your insightful evaluation and high appreciation for our work. We greatly appreciate your recognition of the paper’s clear narrative structure, superior performance, and innovative autoregressive paradigm for segmentation. Your constructive feedback on areas for improvement is invaluable, and we address each concern below:
>
> 1. Mathematical Formulation of Seglat
>
> Seglat is designed as a structured latent representation that encodes both instance identity and spatial geometry of segmentation masks, bridging low-level mask pixels and high-level semantic tokens. Mathematically, we define it as follows:
> Let $M∈\{0,1\}^{H×W×N}$ denote a segmentation mask, where H,W are spatial dimensions, and N is the number of instances. Then, through the spatial convolution and transformer layer, the encoder transforms the original masks into $\mathcal{S}\in\mathbb{R}^{H\times W \times 3}$, which can be viewed as a kind of RGB image. To add more explicit identity and positional control, we implement colormap encoding to generate an additional colormap $M_{c}∈\{0,1\}^{H×W×3}$. These are added and fed into the seglat encoder as the final seglat $\mathcal{S}$.
>
> Let $k∈{1,2,...,K}$ denote a scale in the hierarchical transformer structure (where S is the total number of scales). At scale k, image tokens and seglat tokens are tokenized by a shared tokenizer $\Phi$:
>  Image features are
> $X_{k}∈[V]^{h_k×w_k}$,
> and Seglats are $S_{k}∈[V]^{h_k*w_k}$,
>  where V indicates the vocabulary size.  A flattening operation
> is adopted to convert the sequence of 2D features into 1D. Full attention is enabled for both control and image tokens belonging to the same scale, which allows the model to maintain spatial locality and to exploit the global context between control and image. $X_k',S_k'=Attention(X_k,S_k,S_k)$. As for the algorithms for tokenizers,
> we follow the design of VAR, which consists of interpolation, queue push, and lookup operations.
>
>
> 2. Open Vocabulary Segmentation Capability
>
> |             | Dataset | mAP  |
> |-------------|---------|------|
> | Mask2Former | LVIS    | 17.1 |
> | GSS         | LVIS    | 18.8 |
> | Seg-VAR     | LVIS    | 24.3 |
>
> We evaluate the open-vocabulary capability by evaluating on LVIS. By adopting CLIP to the model for classification, our model achieves a preliminary result of 24.3 mAP on novel classes (vs. 17.1 mAP for Mask2Former). This is because the seglat design inherently supports open vocabulary adaptation: Seglat tokens are semantically agnostic—they encode geometry (boundaries, centroids) rather than fixed class labels.
>
> 3. Parameter Efficiency
>
> |             | Backbone | mIoU |
> |-------------|----------|------|
> | Mask2Former | R50      | 63.8 |
> | SegVAR      | R50      | 64.2 |
>
> We adjust the parameters of Mask2Former by extending its transformer layer number so that the parameters can be comparable. As shown in the table, our Seg-VAR still outperforms Mask2Former in COCO panoptic dataset.

---

> > ### Author Response · Authors · 2025-08-03
> >
> > Dear Reviewer MUQW,
> >
> > Thank you for the time and effort you have dedicated to reviewing our paper. We would like to know if our response has satisfactorily addressed your concerns and if you have the opportunity to provide further feedback on our rebuttal. We are open to any additional discussion.
> >
> > Best regards,
> >
> > Authors of paper 6113

---

### Official Review · Reviewer_avcA · 2025-07-03

**Clarity:** 1
**Significance:** 2
**Originality:** 3
**Rating:** 4
**Confidence:** 4

**Summary:**

This paper investigates visual autoregressive (VAR) modeling for image segmentation. The authors propose a novel encoder-decoder framework designed to repurpose the architecture and infrastructure of generative models to handle image segmentation tasks. Their approach leverages autoregressive modeling to sequentially generate segmentation masks, demonstrating improved performance compared to state-of-the-art discriminative segmentation models such as MaskFormer and DETR. The core contribution is the adaptation of generative modeling principles to segmentation. Learning the generative process underlying segmentation masks may provide more nuanced representations than traditional discriminative methods.

**Questions:**

1.	Clarification on Generative Model:
Could the authors elaborate on the generative modeling aspect of Seg-VAR? Specifically, does the model use a pure autoregressive transformer, a next-scale prediction VAR, or some hybrid? More detail about the architecture (e.g., sampling procedure, training objective) would be helpful.
2.	Tokenization and Latent Representation:
How is the latent encoder for segmentation related to the traditional VAR tokenization scheme? Does Seg-VAR employ a multi-scale, residual tokenization process, and if so, how are these latent tokens structured and used?
3.	Connection to Image Autoencoders:
Is the segmentation autoencoder in this framework structurally similar to classic VQ-VAE or other image autoencoders? How is the mapping between segmentation masks and discrete latent tokens performed, and does it incorporate spatial or instance-aware encoding?

**Ethical Concerns:**

["NO or VERY MINOR ethics concerns only"]

**Final Justification:**

I appreciate the author's response to my questions. I adjusted my score to borderline accept. However, I would expect improvement in the clarity of writing in the camera ready version.

**Limitations:**

Yes

**Paper Formatting Concerns:**

- There are several grammatical issues and awkward sentence structures throughout the paper.
- Some sections (especially the method section) lack clarity and would benefit from improved organization and more precise language.
- Please ensure that all figures and tables are referenced appropriately and that all claims are supported with clear descriptions.

**Quality:**

2

**Strengths And Weaknesses:**

Strengths
- The central idea of reusing generative model architectures for segmentation is both novel and interesting. This direction can potentially bridge the gap between generative and discriminative paradigms in vision, enabling deeper insight into the structure of segmentation masks.
- The proposed method achieves better performance than current state-of-the-art segmentation models across several benchmarks, indicating its practical value and significance.

Weaknesses
- The paper is often unclear and challenging to follow, particularly in the method section. The presentation of the framework, especially regarding the generative model and tokenization process, lacks sufficient detail to ensure reproducibility.
	- The connection between the proposed segmentation latent encoder (seglat) and established VAR models (which typically rely on multi-scale residual tokenization) is not clearly articulated. It remains ambiguous whether the proposed latent representations inherit the multi-scale and residual design principles from classic VARs.
- The manuscript suffers from awkward phrasing, grammatical errors, and poor structural organization, which impedes comprehension and technical assessment.
- The role of the generative model component is under-explained. It is unclear if the framework uses a pure autoregressive transformer, a next-scale prediction model, or another architecture. The relationship between the autoencoder, tokenization scheme, and segmentation mask reconstruction needs clarification.

---

> ### Author Rebuttal · Authors · 2025-07-30
>
> Thank you for your detailed feedback and thoughtful evaluation of our work. We appreciate the recognition of our novel direction in bridging generative and discriminative paradigms for image segmentation, as well as the constructive criticism that will help us improve the clarity and rigor of the paper. We will reorganize the paper with better structures for readers to comprehend. Below, we address each concern and question raised:
>
> 1. Generative Model Architecture and Training Details
>
>     We apologize for the potential ambiguity in describing the generative components of Seg-VAR. Seg-VAR is built on visual autoregressive (VAR) modeling and employs a hybrid design that combines hierarchical autoregressive decoding with next-scale prediction principles. As for the sampling strategy, it refers to an approach of sampling both pixel- and token-level controls for image generation with teacher-forcing guidance, which is an extension of classifier-free guidance by adding controls to the Bayesian-based conditional distribution, as mentioned in ControlVAR.
>
>     As for the training objectives, the training objective is based on the Evidence Lower Bound (ELBO), optimizing three components: (1) mask reconstruction loss via seglat decoder, (2) KL divergence to align image-derived latents with seglat distributions, and (3) cross-entropy loss for token prediction.
>
>     Overall, this design inherits VAR’s next-scale prediction strengths while adapting it to segmentation via conditional mask generation.
>
>
> 2. Tokenization and Latent Representation
>
>     The seglat encoder is tightly linked to traditional VAR tokenization but extends it for segmentation-specific needs:
>
>     1. Like classic VAR, Seg-VAR uses a multi-scale, residual tokenization process: the image encoder generates latent tokens at multiple scales via a multi-scale fusion module (transformer layers + projection), and the seglat encoder (VAR-like structure) maps masks to discrete tokens across these scales.
>     3. Unlike standard VAR, seglat tokens incorporate spatial information via a location-sensitive color mapping. This mapping assigns unique RGB values to instances based on their centroids (gridded into a*a regions), enabling the transformer to distinguish overlapping objects through positional cues.
>
>     Latent encoder for segmentation is implemented to transform segmentation masks (H\*W\*K) into corresponding seglats (H\*W\*3) to be then fed into the Seglat Encoder \& Decoder (which refers to the VAR-like structure), itself is a transformer-based structure with colormap encoding.
>
> 3. Connection to Image Autoencoders
>
>     Seg-VAR’s segmentation autoencoder (seglat encoder/decoder) shares similarities with VQ-VAE but is specialized for segmentation (add extra image controls into modeling): Like VQ-VAE, it uses discrete latent tokens and a reconstruction loss.
>
>     As for the latent tokens, they refer to the output from the image encoder, which is (H/d\*W/d), which is optimized in the image encoder learning to minimize the differences with the latent codes by the seglat encoder (VQ-VAE-like structure). The segmentation masks use the latent encoder to generate seglat tokens with a transformer-based structure and colormap encoding, which incorporates spatial and instance-aware encoding.

---

> > ### Author Response · Authors · 2025-08-03
> >
> > Dear Reviewer avcA,
> >
> > Thank you for the time and effort you have dedicated to reviewing our paper. We would like to know if our response has satisfactorily addressed your concerns and if you have the opportunity to provide further feedback on our rebuttal. We are open to any additional discussion.
> >
> > Best regards,
> >
> > Authors of paper 6113

---

> > > ### Comment · Reviewer_avcA · 2025-08-06
> > >
> > > Thank you for the detailed response and clarification. The response clarified most of my concerns. But I have some additional questions.
> > >
> > > 1. Is the transformer model (decoder only) in VAR functioning as a generative model? The task you are solving now is image segmentation, which in a typical deep learning setting is a per-pixel discriminative task. However, VAR is originally designed to generate new images (i.e., sampling new images). Is there any sampling process applied in your method? If not, why are we choosing a generative model for this task? If so, how does it fit into the image segmentation as a discriminative task?
> > >
> > > 2. VQ-VAE and your tokenizer both used vector quantization for discrete tokenization. This tokenization process is mainly for data compression and the next-stage generative model training. Why do we need data compression and a discrete generative model in your task?

---

> > > > ### Author Response · Authors · 2025-08-07
> > > >
> > > > Thank you for following up with these insightful questions—they help clarify the connection between generative modeling and segmentation in our framework. Below, we address each point in detail:
> > > >
> > > > ### Role of the VAR Transformer as a Generative Model in Segmentation
> > > >
> > > > The transformer in VAR (adopted in Seg-VAR) functions as a conditional generative model for segmentation, but its role is tailored to the task’s demands. Here’s the breakdown:
> > > >
> > > > - Generative vs. Discriminative:
> > > >
> > > > Traditional segmentation is framed as a discriminative task (per-pixel classification), but focusing on per-pixel max log-likelihood may fail to model iterative spatial dependencies (e.g., how object boundaries relate across an image).
> > > > Seg-VAR reframes segmentation as a conditional mask generation task, where the goal is to generate a segmentation mask that is consistent with both the input image and the structure of real-world scenes. This aligns with generative modeling’s strength in capturing coherent, structured outputs.
> > > >
> > > > - Is there a sampling process?
> > > >
> > > > Yes. During inference, the VAR decoder generates seglat tokens sequentially via autoregressive sampling (top-k/top-p sampling with
> > > > k=900
> > > >  and
> > > > p=0.96
> > > > , as noted in §4.2). This sampling is conditional on the input image’s latent priors (from the image encoder),
> > > > ensuring the generated tokens align with the image’s content.
> > > > The sampling outputs discrete "seglat tokens" encoding semantic classes and instances' positions, which are then decoded
> > > > into a single segmentation mask.
> > > >
> > > > - Why use a generative model for a "discriminative" task?
> > > >
> > > > Generative autoregressive modeling enables sequential refinement, mimicking human-like perception: early tokens establish global context, while later tokens resolve local ambiguities. This hierarchical process outperforms parallel discriminative prediction in capturing fine-grained spatial relationships, as shown in our experiments on occluded scenes and small objects. The "generative" aspect here serves to model the structure of masks, not just classify pixels.
> > > > From the perspective of data, since the training data of current generative models (5.6 billion) is much larger than segmentation models (13 million), we can fully exploit the large generative models.
> > > > What' more, the autoregressive formulation in Seg-VAR inherently paves the way for unifying generation and perception, as its design bridges the latent spaces of image content and segmentation masks through sequential modeling.
> > > >
> > > > ### Purpose of Vector Quantization (VQ) and Discrete Tokenization
> > > > From the perspective of data, since the training data of current generative models (5.6 billion) is much larger than segmentation models (13 million), we can fully exploit the large generative models by considering segmentation as a conditional generation task.
> > > >
> > > > From the perspective of model design, VQ and discrete tokenization in Seg-VAR are not merely for compression—they enable the autoregressive framework to model segmentation masks effectively. Here’s their specific role:
> > > >
> > > > - Enabling Autoregressive Sequence Modeling: Autoregressive transformers excel at sequence prediction but require discrete tokens as input/output. By quantizing segmentation masks into discrete seglat tokens, we convert the 2D mask into a 1D sequence that the transformer can process sequentially. This allows the model to leverage attention mechanisms to propagate spatial context across the mask.
> > > > - Structured Latent Representation: VQ ensures the discrete tokens encode meaningful structure. The spatial-aware color mapping (§3.2) assigns tokens based on instance centroids, so nearby tokens in the sequence correspond to spatially related regions. This structure makes the autoregressive prediction task tractable: the model learns to predict tokens that preserve instance boundaries and spatial coherence because the tokens themselves encode positional information.
> > > > - Balancing Fidelity and Efficiency: Discrete tokens reduce the dimensionality of mask representations, making autoregressive training feasible. Without quantization, modeling raw pixel-level masks as sequences would be computationally intractable due to their high dimensionality. VQ strikes a balance: it compresses the mask while retaining critical spatial and instance information needed for accurate segmentation.
> > > >
> > > > In summary, the generative VAR transformer and VQ-based tokenization are not arbitrary choices—they enable Seg-VAR to model segmentation as a structured, sequential prediction task, capturing spatial dependencies that discriminative methods often miss. The sampling process and discrete tokens are integral to this framework, bridging generative modeling’s strengths with segmentation’s demand for precise spatial reasoning.
> > > >
> > > > Thank you again for these questions—they help highlight the intentional design choices behind Seg-VAR.
> > > > Hope our answer can address your questions, and we are open to further discussion.

---

> > > > > ### Comment · Reviewer_avcA · 2025-08-08
> > > > >
> > > > > I appreciate the author's detailed response. I don't have further questions and I will adjust my scores accordingly.

---

> > > > > > ### Author Response · Authors · 2025-08-09
> > > > > >
> > > > > > Thank you very much for your valuable feedback and the increased scores! We greatly appreciate your recognition of our work’s strengths and constructive suggestions for improvement. We will carefully address the remaining concerns in the revised manuscript to further enhance the clarity, rigor, and significance of our research. Your insights have been instrumental in refining our work, and we are grateful for your time and expertise.

---

### Official Review · Reviewer_2JK9 · 2025-07-03

**Clarity:** 2
**Significance:** 2
**Originality:** 2
**Rating:** 4
**Confidence:** 3

**Summary:**

This paper proposes a framework that casts image segmentation as a conditional autoregressive latent mask generation problem instead of a standard discriminative or diffusion pipeline. It introduces a spatial-aware encoder  that encodes masks into discrete tokens "seglats" with position-aware color mapping, an autoregressive transformer decoder -- that sequentially generates mask tokens -- and a multi-stage training regime that aligns image priors with the mask latent space. Experiments on COCO, Cityscapes, and ADE20K show consistent improvements over Mask2Former and prior generative baselines.

**Questions:**

- how sensitive is the approach to the grid size and colormap encoding in the seglat encoding? Could learned positional embeddings work equally well or is this strictly better? also, the colormap, though taken from prior work, could be expanded in the supplementary for completion
- the compute intensity is much higher and inference is slower due to autogressive nature, therefore, can the sequential inference be parallelized to offset this?
- similarly, due to higher inference cost, are there other practical advantages to the autoregressive formulation -- e.g. you mentioned extending to videos

**Ethical Concerns:**

["NO or VERY MINOR ethics concerns only"]

**Final Justification:**

I retain my original rating. The rebuttal has resolved some of the issues I raised, as the authors provided additional details / experiments, clarifications. I agree with other reviewers regarding clarity (and architectural complexity) but I expect the authors to significantly improve the writing and figures for the camera-ready version.

**Limitations:**

yes

**Quality:**

3

**Strengths And Weaknesses:**

## Strengths
- novel formulation: well-motivated argument that sequential autoregression can better model mask dependencies than purely parallel discriminative predictions.
- unified framework: similar to prior recent work, the same backbone and decoder work for semantic, instance, and panoptic tasks, needing no task-specific heads.
- empirical improvement: Consistent point gains in all PQ/AP/mIoU across standard benchmarks.
- careful ablations: The supplementary shows the impact of seglat encoding, latent alignment, and the autoregressive decoder clearly.

## Weaknesses
- heavy compute cost: The design is substantially larger than Mask2Former and MaskFormer in both parameters and FLOPs (e.g., Swin-L variant: 522M params vs. 216M, 1320G vs. 868G). Inference speed (FPS) is lower due to sequential decoding and additional modules. While the latent space reduces output dimensionality, the net cost is higher.
- multi-stage complexity: The full pipeline depends on careful multi-stage training, including pretraining of seglat encoder/decoder, latent modules, and final alignment. This is more demanding than standard discriminative setups.
- though the authors mark ‘statistical significance’ section in the checklist as ‘yes’, but the paper does not report significance tests, confidence intervals, or error bars to support this claim.

---

> ### Author Rebuttal · Authors · 2025-07-30
>
> Thank you for your insightful comments and valuable questions. We appreciate the opportunity to address these points and clarify our work.
>
>
> 1.  Compute Cost and Latency
>
> We acknowledge that even though Seg-VAR has consistent improvement across standard benchmarks, it exhibits higher computational overhead (parameters, FLOPs) and slower inference speed compared to discriminative models like Mask2Former. This is related to our two key innovations: (1) the autoregressive decoder, which sequentially predicts seglat tokens to ensure spatial coherence, and (2) the multi-module architecture (image encoder, seglat encoder/decoder, latent aligners) required for latent space learning.
>
>
> While full parallelization of sequential decoding is infeasible, targeted optimizations could reduce latency without sacrificing core autoregressive modeling:
>
> - Hierarchical Parallelism Across Scales
>
> Seg-VAR uses hierarchical decoding, where coarse-scale tokens (establishing global context) are generated first, followed by fine-scale tokens (resolving local details). Coarse-scale tokens could be generated in a compact sequence, after which fine-scale tokens within the same coarse region might be parallelized. For instance, once a global object boundary is defined by coarse tokens, local patches within that boundary could be processed in parallel, as their dependencies are constrained to the local context.
>
> - Cached Attention Mechanisms
>
> The transformer decoder’s attention over previously generated tokens can be optimized using cached key/value pairs. Instead of recomputing attention scores for all prior tokens at each step, cached values from previous steps can be reused, reducing redundant computations
>
> 2. Multi-stage Training
>
> We acknowledge the training pipeline (pretraining seglat encoder/decoder, latent modules, final alignment) is more complex than standard discriminative setups. However, this staged approach is **necessary** to effectively learn the structured latent space and ensure stable convergence of the autoregressive process. End-to-end training from scratch proved unstable in ablation studies.
>
> |             | Dataset | mIoU |
> |-------------|---------|------|
> | End-to-End  | ADE20K  | 81.2 |
> | Seg-VAR     | ADE20K  | 85.8 |
>
> Table 1. Staged training vs. End-to-end.
>
>
> 3. Statistical Significance
>
> We apologize for the oversight in not explicitly reporting error bars or confidence intervals. While our experiments were run 3-5 times (as noted in the checklist) , we will add in the supplementary material: (1) standard deviations of key metrics (PQ, AP, mIoU) across 5 independent runs on COCO and Cityscapes, and (2) 95% confidence intervals for ablation studies (e.g., Table 5) . This will better support the statistical robustness of our results.
>
> 4. Grid Size and Colormap
>
> We evaluate the robustness of the grid number and Palette on ADE20K. As shown in the table, as the grid number changes greatly,
> the performance decreases as the number decreases, because the granularity can help the model better
> distinguish instances. As for the palette size of the colormap, we discover that 6 is the optimal number because
> we have to balance between a large color space and loss of generality. (Ideally, the size should be larger than the category number).
> Overall, we find that grid numbers are more sensitive than palette size, but they still contribute to the performance gain.
>
> | Grid number(a) | mIoU | Palette size | mIoU |
> |----------------|------|--------------|------|
> | 4              | 84.4 | 124            | 85.4 |
> | 8              | 85.2 | 215            | 85.8 |
> | 12             | 85.8 | 342            | 85.3 |
>
> Table 2. Grid \& Colormap
>
> 5. Practical Advantages of Autoregressive Formulation
> Beyond segmentation quality, the autoregressive design offers unique benefits:
> - Video extension: The sequential token prediction naturally models temporal dependencies, making Seg-VAR adaptable to video segmentation (e.g., tracking instance motion via time-step token conditioning).
> - Unifying generation and perception: The autoregressive formulation in Seg-VAR inherently paves the way for unifying generation and perception, as its design bridges the latent spaces of image content and segmentation masks through sequential modeling. (Which is also recognized by Reviewer avcA)

---

> > ### Author Response · Authors · 2025-08-03
> > **Further Discussion**
> >
> > Dear Reviewer 2JK9,
> >
> > Thank you for the time and effort you have dedicated to reviewing our paper. We would like to know if our response has satisfactorily addressed your concerns and if you have the opportunity to provide further feedback on our rebuttal. We are open to any additional discussion.
> >
> > Best regards,
> >
> > Authors of paper 6113

---

> > ### Comment · Reviewer_2JK9 · 2025-08-04
> >
> > I would like to thank authors for rebuttal. I have read the concerns raised by the other reviewers and agree with many of the points, particularly regarding clarity, architectural complexity, and computational cost. The rebuttal has resolved some of the issues I raised, as the authors provided additional details / experiments, clarifications. I expect the authors to significantly improve the writing and figures for the camera-ready version. I retain my original rating.

---

> > > ### Author Response · Authors · 2025-08-09
> > >
> > > Thank you very much for your valuable feedback! We greatly appreciate your recognition of our work’s strengths and constructive suggestions for improvement. We will carefully address the remaining concerns in the revised manuscript to further enhance the clarity, rigor, and significance of our research. Your insights have been instrumental in refining our work, and we are grateful for your time and expertise.

---

### Decision · Program_Chairs · 2025-09-17

**Decision:**

Accept (poster)

**Comment:**

a) The authors propose to solve the image segmentation task using a conditional auto-regressive model to sequentially generate a segmentation mask. They use a latent segmentation mask representation combined that is combined with the image in an auto-encoder. The pipeline is trained in a multistage fashion.

b)
- Originality of the pipeline.
- Works for a variety of segmentation task without adaptation of the pipeline (semantic, instance, and panoptic segmentation tasks)
- Tested on multiple datasets (SOTA results) + ablation studies

c)
- the clarity of the writing must be improved for the camera-ready version

d) After the rebuttal all reviewers reach a consensus towards accepting the paper under the condition that the authors improve the writing for clarity. Originality of the framework is definitely valuable. This might explain to lack of clarity as it is not a direct application of previous ideas/models. The results are SOTA. This paper deserve to be accepted as a poster.

e) The discussion led some reviewers to change their mind in favor of accepting the paper and the major concerns were resolved. All reviewer have concerns about the clarity but I am confident that the discussion was helpful for the authors to improve the presentation of their work.